# Large-Scale International Validation of an Indirect ELISA Based on Recombinant Nucleocapsid Protein of Rift Valley Fever Virus for the Detection of IgG Antibody in Domestic Ruminants

**DOI:** 10.3390/v13081651

**Published:** 2021-08-19

**Authors:** Janusz T. Pawęska, Petrus Jansen van Vuren, Veerle Msimang, Modu Moustapha Lô, Yaya Thiongane, Leopold K. Mulumba-Mfumu, Alqadasi Mansoor, José M. Fafetine, Joseph W. Magona, Hiver Boussini, Barbara Bażanow, William C. Wilson, Michel Pepin, Hermann Unger, Gerrit Viljoen

**Affiliations:** 1Centre for Emerging Zoonotic and Parasitic Diseases, National Institute for Communicable Diseases of the National Health Laboratory Service, Sandringham 2131, South Africa; veerlem@nicd.ac.za; 2Centre for Viral Zoonoses, Department of Medical Virology, Faculty of Health Sciences, University of Pretoria, Pretoria 0001, South Africa; 3Faculty of Health Sciences, School of Pathology, University of Witwatersrand, Johannesburg 2050, South Africa; 4Australian Centre for Disease Preparedness, CSIRO Health & Biosecurity, Geelong, VIC 3220, Australia; Petrus.Jansenvanvuren@csiro.au; 5Laboratoire National de l’Elevage et de Recherches Vétérinaires, Route de Front de Terre, Dakar Hann 2057, BP, Senegal; moustaphlo@yahoo.fr (M.M.L.); yayathiongane@hotmail.fr (Y.T.); 6Ministry of Agriculture, Democratic Republic of Congo, Kinshasa 7948, Democratic Republic of the Congo; leopold_mulumba@yahoo.com; 7Department of Clinical Sciences, Faculty of Veterinary Medicine, University of Kinshasa, P.O. Box 127, Kinshasa XI, Democratic Republic of the Congo; 8Central Veterinary Laboratory, General Directorate of Animal Health & Veterinary Quarantine, Ministry of Agriculture and Irrigation, Sana’a 31220, Yemen; Mansoor.Alqadasi@fao.org; 9Food and Agriculture Organization Office, Sana’a 31220, Yemen; 10Veterinary Faculty, Eduardo Mondlane University, Maputo 1103, Mozambique; jfafetine153@gmail.com; 11National Livestock Resources Research Institute, Tororo P.O. Box 96, Uganda; magona.joseph@gmail.com; 12Food and Agriculture Organization, Gaborone P.O. Box 54, Botswana; 13Direction Generale Des Services Veterinaires, Ministère des Ressources Animales, Ouagadougou 09 BP 907, Burkina Faso; hiver.boussini@au-ibar.org; 14African Union Interafrican Bureau for Animal Resources, Nairobi P.O. Box 30786-00100, Kenya; 15Department of Pathology, Faculty of Veterinary Science, University of Environmental and Life Sciences, 50-375 Wroclaw, Poland; barbara.bazanow@upwr.edu.pl; 16United States Department of Agriculture, Agricultural Research Service, Foreign Arthropod Borne Animal Diseases Research Unit, National Bio- and Agro-Defense Facility, Manhattan, KS 66502, USA; william.wilson2@usda.gov; 17Agence Française de Sécurité Sanitaire des Aliments, F-69364 Lyon, France; michel.pepin@vetagro-sup.fr; 18VetAgro Sup, Campus Vétérinaire de Lyon, F-69364 Lyon, France; 19Joint FAO/IAEA Centre for Nuclear Techniques in Food and Agriculture, International Atomic Energy Agency, 1400 Vienna, Austria; ungervet@gmail.com (H.U.); G.J.Viljoen@iaea.org (G.V.)

**Keywords:** Rift Valley fever virus, enzyme-linked immunosorbent assay, recombinant nucleocapsid, IgG antibody, domestic ruminants, validation, diagnostic accuracy

## Abstract

Diagnostic performance of an indirect enzyme-linked immunosorbent assay (I-ELISA) based on a recombinant nucleocapsid protein (rNP) of the Rift Valley fever virus (RVFV) was validated for the detection of the IgG antibody in sheep (*n* = 3367), goat (*n* = 2632), and cattle (*n* = 3819) sera. Validation data sets were dichotomized according to the results of a virus neutralization test in sera obtained from RVF-endemic (Burkina Faso, Democratic Republic of Congo, Mozambique, Senegal, Uganda, and Yemen) and RVF-free countries (France, Poland, and the USA). Cut-off values were defined using the two-graph receiver operating characteristic analysis. Estimates of the diagnostic specificity of the RVFV rNP I-ELISA in animals from RVF-endemic countries ranged from 98.6% (cattle) to 99.5% (sheep) while in those originating from RVF-free countries, they ranged from 97.7% (sheep) to 98.1% (goats). Estimates of the diagnostic sensitivity in ruminants from RVF-endemic countries ranged from 90.7% (cattle) to 100% (goats). The results of this large-scale international validation study demonstrate the high diagnostic accuracy of the RVFV rNP I-ELISA. Standard incubation and inactivation procedures evaluated did not have an adverse effect on the detectable levels of the anti-RVFV IgG in ruminant sera and thus, together with recombinant antigen-based I-ELISA, provide a simple, safe, and robust diagnostic platform that can be automated and carried out outside expensive bio-containment facilities. These advantages are particularly important for less-resourced countries where there is a need to accelerate and improve RVF surveillance and research on epidemiology as well as to advance disease control measures.

## 1. Introduction

The geographic expansion of Rift Valley fever virus (RVFV) in the last four decades associated with high health and socio-economic losses is of great concern for veterinary and public health worldwide. The wide distribution of potentially competent mosquito vectors in different geographic regions of the world and increased international trade and travel carry the risk of the introduction and spread of this zoonotic virus to RVF-free areas [1,2,3,4]. The unpredictable and sudden emergence of RVFV outside traditional endemic areas, unavailability of safe and efficacious antiviral treatment, and prophylactic immunization led the World Health Organization (WHO) to recognize RVF as a priority disease for the development of accurate diagnostics, effective therapeutics, and vaccines [5].

Clinical manifestations of Rift Valley fever (RVF) in livestock vary between species and depend largely on the age of the infected animal. Most severe symptoms are seen in small ruminants, where so-called “abortion storms” may result in very high fetal and neonatal losses [6,7]. Clinical signs in humans vary from mild flu-like conditions to meningoencephalitis, retinitis, and hemorrhagic fever syndrome [8,9]. RVFV is suspected to induce miscarriages in women [10]. RVFV belongs to a group of viral hemorrhagic fever (VHF) agents regarded as a potential bioweapon with high adverse impacts on public health and agriculture [11,12]. As for most VHFs, the non-specific presentation of RVF makes it difficult to diagnose clinically. Therefore, the differential diagnosis in both humans and animals concerns a broad array of conditions, especially when first cases are encountered during a yet unrecognized outbreak. RVF may be suspected when there is a sudden outbreak of febrile illness with headache and myalgia in humans, in association with the occurrence of abortions in domestic ruminants and deaths of young animals following heavy rains [3,6,9]. RVFV is transmitted among animals mostly by aedine and culicine mosquitoes. Current data suggest that over 50 mosquito species, many of which have global distribution, can potentially act as vectors of RVFV [13,14]. Humans usually become infected following contact with virus-contaminated tissues and body fluids from infected animals, but mosquito bites can also transmit the virus [15,16,17].

RVFV is a negative-stranded RNA virus, a member of the genus *Phlebovirus*, family *Phenuiviridae.* The genome of RVFV comprises three segments, encoding the RNA-dependent RNA polymerase (L segment), the two surface proteins Gn and Gc as well as the nonstructural protein NSm (M segment), the nucleoprotein (NP), and a further nonstructural protein NSs (S-segment) [18]. The N protein is the most abundant protein in phlebovirus-infected cells and strongly immunogenic [19,20].

Various diagnostic methods are available for laboratory confirmation of infections with RVFV. Isolation of RVFV is achieved in hamsters, infant or adult mice, and various cell cultures [6,21]. Highly sensitive genetic amplification assays for the detection and quantification of RVFV in serum and other tissues of infected humans and livestock, as well as mosquitoes, have been reported. These assays include polymerase chain reaction (PCR) [22,23,24,25], real-time RT PCR (RTD-PCR) based on TaqMan probe technology [26,27], and the real-time reverse-transcription loop-mediated isothermal amplification assay [28]. A viral antigen can be detected in blood and other tissues by a variety of immunological methods, including agar gel immunodiffusion and immunostaining assays [6,14,29]. A sandwich ELISA was developed for the detection of the nucleocapsid protein of RVFV in various clinical specimens [30]. The lateral flow immunochromatographic test for the detection of RVFV NP in animal sera and fluids from aborted fetuses provides a valuable diagnostic tool for onsite rapid detection of the virus [31].

Although viremia in infected individuals reaches high titers, it is of short duration, thus limiting the use of viral antigen and molecular detection systems [13]. The collection of diagnostic specimens after viral clearance and inappropriate transportation methods and storage conditions may negatively affect molecular assays, making serology testing an important diagnostic capacity in the veterinary and public health response to outbreaks occurring in remote locations where limited resources are available. Most adult animals and infected patients undergo subclinical or mild infections; therefore, antigen and nucleic acid detection assays should be run in parallel with antibody-detecting techniques. Type-specific antibodies to RVFV are easily demonstrable shortly after exposure to the virus. Serodiagnosis of recent infection can be confirmed by demonstrating seroconversion or a fourfold or greater rise in titer of the antibody in paired serum samples or by the detection of IgM antibody [9,13].

The classical methods for the detection of antibodies to RVFV include hemagglutination inhibition, complement fixation, indirect immunofluorescence, and virus neutralization tests [6,14,32]. Although regarded as a gold standard, the virus neutralization tests are laborious, expensive, and require 5–7 days for completion. These assays can be performed only when a standardized stock of live virus and tissue cultures are available, which poses a health risk to laboratory personnel [33,34], thus restricting their use outside RVF-endemic areas and/or to high bio-containment facilities. Consequently, virus neutralization assays are rarely used, and then only in highly specialized reference laboratories [14]. To address this issue, a virus neutralization test based on an avirulent RVFV expressing an enhanced green fluorescent protein was developed and reported to be more sensitive than the classical neutralization test [35]; however, this test takes multiple days to perform.

Various enzyme-linked immunosorbent assay (ELISA) formats have been developed and validated in recent years for the specific detection of anti-RVFV antibodies in humans and animals, based on β-propiolactone inactivated or gamma-irradiated antigens derived from infected tissue cultures or mouse liver [36,37,38,39,40,41]. An optical fiber immunosensor based on a sandwich ELISA using a gamma-irradiated RVFV whole antigen was developed for the detection of the anti-RVFV IgG antibody in human sera [42]. While ELISAs based on an inactivated whole antigen of RVFV have high diagnostic accuracy compared to virus neutralization assays [36,37,38,39,40,41], the production of antigens for these assays also requires high bio-containment facilities. Recombinant antigen technology allows the production of high-quality viral antigens under biosafety level two conditions. Indirect ELISAs based on the recombinant NP or GP proteins for the detection of anti-RVFV antibodies have been developed [43,44,45,46]. An ELISA platform based on recombinant NP and NSs proteins can distinguish infected from vaccinated animals. Given very strict regulations for the international trade of animals from RVF-endemic countries, this diagnostic tool has the potential to assist in the safe movement and trade of domestic ruminates [47]. A multiplex fluorescence microsphere immunoassay (FMIA) was developed to detect IgM and IgG antibodies in ruminant sera to RVFV structural and non-structural proteins. Preliminary results demonstrate the potential of the FMIA diagnostic platform for the development of diagnostic tests that can be used to differentiate vaccinated from infected animals and for simultaneous differential diagnosis of several abortive and zoonotic pathogens [48,49].

The inhibition ELISA based on a whole tissue culture-derived, inactivated antigen [40] and competitive ELISA based on recombinant NP antigen [50,51,52] allow multi-species RVFV antibody detection using the same diagnostic procedure without requirements for species-specific conjugates. While in the last 15 years, several groups have developed and evaluated various ELISA formats based on RVFV recombinant NP antigen [43,44,45,46,47,48,49,50,51,52,53,54,55,56,57,58,59,60], more extensive evaluation of their diagnostic performance was only achieved in humans [58], buffalo [59], and cattle [60] to date.

The increasing importance of RVF as a zoonotic threat and the need for a better understanding of RVF epidemiology requires the application of well-evaluated tools for antibody detection across different geographic areas and domestic and wildlife ruminants implicated in RVF epidemiology. Reasons for the test validation include the need for a statistically sound evaluation of assay diagnostic performance parameters that are essential for reporting diagnostic results, comparing results between different diagnostic laboratories, determining seroprevalence rates, designing infection risk population studies, and assessing the occurrence of asymptomatic infections. The process of an ELISA standardization and validation is, however, complex, time-consuming, expensive, and vulnerable to many limitations, including the availability of recommended standards and representatives of the reference sera [61,62,63,64].

In response to the increasing demand for safe serological diagnosis of RVF, we validated the diagnostic performance of RVFV rNp-based I-ELISA for the detection of IgG antibodies using large serum panels collected from sheep, goats, and cattle in RVF-endemic and RVF-free countries.

## 2. Material and Methods

### 2.1. Origin of Serum Specimens

A total of 9818 individual banked sera from sheep (*n* = 3367), goats (*n* = 2632), and cattle (*n* = 3819), collected in RVF-endemic (*n* = 6810) and RVF-free countries (*n* = 3008) in 2000–2009, were used (Table 1). These sera represent submissions to national veterinary laboratories for routine diagnostic testing. They were shipped to South Africa on ice packs under veterinary import permits for the importation of diagnostic specimens, issued by the Directorate of Animal Health, Department of Agriculture, South Africa (Permit Nos.: 13/1/30/4-131 and 13/1/1/30/2/1/20-341). Upon shipment, sera were stored at −70 °C at the National Institute for Communicable Diseases until tested. The RVF vaccination or infectious status of sampled animals was unknown.

### 2.2. VNT

A virus neutralization test (VNT) was used to dichotomize sera according to RVFV infection status. Each serum was tested in duplicate in VNT as described previously using the AR 20368 strain of RVFV isolated in 1981 from *Culex zombaensis* in South Africa [38]. Titers were expressed as the reciprocal of the serum dilution that inhibited ≥75% of the viral cytopathic effect. A serum sample was considered positive for anti-RVFV neutralizing antibodies when it had a titer of ≥1:10.

### 2.3. rNP I- ELISA

Previously produced freeze-dried, gamma-irradiated sheep serum controls were used [38,58]. The production of rNP antigen for I-ELISA in *E. coli* expression system and the assay procedure were carried out as described previously [44] with minor modifications. The NP sequence is based on the Zim688/78 RVFV strain isolated from a bovine in 1978 in Zimbabwe [44]. The coding sequence is available in Genbank, accession number DQ924959. Briefly, all control and test sera and reagents were added to microtiter 96-wells plates (MaxiSorb Immunoplates, Nunc, Roskild, Denmark) at a volume of 100 µL/well unless otherwise stated. The passive absorption of stock RVFV rNP antigen was performed at 4 °C overnight in carbonate/bicarbonate buffer (pH 9.6), and all subsequent incubations (except for substrate addition) were performed at 37 °C in a humidified chamber for 1 h. Following coating, plates were washed three times with 300 µL/well of phosphate-buffered saline (PBS) pH 7.2 and 0.1% Tween; the same washing procedure followed each subsequent stage of I-ELISA. Plates were blocked with 200 µL/well 10% fat-free milk powder in PBS. After incubation, plates were washed, and control and test sera diluted 1:400 in PBS containing 2% milk powder (diluent buffer) were added to the plates. Each test serum was assayed in duplicate and each internal control was tested in quadruplicate. After incubation, plates were washed and the HRPO conjugated Protein G (Zymed Laboratories, Inc., South San Francisco, CA, USA), diluted 1:5000 in diluent buffer, was added to the wells. Afterward, incubation plates were washed and 2,2′-azinodiethylbenzothiazoline sulfonic acid peroxidase (Seracare Lifesciences, Milford, MA, USA) substrate was added to wells. Plates were incubated in the dark at room temperature (±22 °C) for 30 min. Reactions were stopped by the addition of 1% sodium dodecyl sulfate, and the optical density readings (OD) were measured at 405 nm. OD readings were converted into a percentage positivity (PP) of high-positive control serum (C++) using the following equation: [PP = (mean test serum OD/mean OD C++) × 100] [36].

### 2.4. Selection of Cut-Off Values

Optimization of rNP I-ELISA cut-off PP values at 95% diagnostic accuracy level were performed using the misclassification costs term (MCT) option [65] of the two-graph receiver operating characteristics (TG-ROC) analysis. TG-ROC is a Microsoft EXCEL spreadsheet template designed for selecting cut-off values in quantitative diagnostic tests at a preselected accuracy level (e.g., 90% or 95% sensitivity and specificity) [66,67]. The optimization of cut-off values was based on the following equation: MCT = (1 − p) (1 − Sp) + rp (1 − Se), where p (prevalence) = 0.5, r (costs of false-positive and false-negative results) = 1, Sp = specificity, and Se = sensitivity [65]. In addition, the cut-off values were determined as the mean plus two standard deviations (mean + 2SD) as well as by the mean plus three standard deviations (mean + 3SD) of the results observed with the RVF-free subpopulations (VNT-negative animals).

### 2.5. Determination of Diagnostic Accuracy and Other Statistical Analysis

Estimates of diagnostic sensitivity (DSn) and diagnostic specificity (DSp) and their 95% confidence intervals (CI), Youden’s index (J), efficiency (Ef), positive predictive value (PPV), and negative predictive value (NPV) were calculated as previously described [39,64]. Means and standard deviations and ranges of I-ELISA PP values of the rNP I-ELISA were calculated by endemicity and by country. Based on the Shapiro–Wilk test that determined if the samples came from a normal distribution, a non-parametric Mann–Whitney U-test was performed to assess whether there was a significant difference between the median PP values of RVF-endemic and RVF-free countries by species of sheep, goats, or cattle. Using Duncan’s test [68] for descriptive statistics, we determined which pairs of means resulting from the country comparison were significantly different from each other by livestock species tested. All calculations and tests were performed using Stata, and a P-value lower than 0.05 was considered statistically significant (Stata Corporation, College Station, TX, USA).

### 2.6. Serum Inactivation

To assess the effect of heat and chemical inactivation on the levels of the detectable anti-RVFV IgG by rNP I-ELISA in ruminant sera, laboratory protocols previously shown to completely inactivate RVFV [30] and other highly hazardous RNA viruses were used [69,70]. The high positive sheep, goat, and cattle sera were first diluted 1:10 in species-corresponding untreated negative serum containing either 0.5% Triton X-100 or 0.5% Tween 20 (Sigma-Aldrich, Taufkichen, Germany) and heated at 60 °C for 15 min. Then, two-fold log_10_ dilutions (from log_10_10^2^ − log_10_10^5.1^) of untreated and treated sera were tested. Each serum dilution was tested in duplicate on 3 separate runs. A serum titer was considered the highest sample dilution at which its PP value was ≥ the rNP I-ELISA cut-off.

### 2.7. rNP I-ELISA Robustness

To assess the robustness of the rNP I-ELISA, the assay was performed using three different incubation conditions: (1) the ELISA plate was coated with rNP overnight at 4 °C, and all subsequent incubations (except for substrate addition) were performed at 37 °C for 1 h (as per the standardized protocol described in Section 2.4); (2) the ELISA plate was coated with rNP at 37 °C for 1 h and all subsequent incubations (except for substrate addition) were carried out at 37 °C for 1 h; (3) all incubations were performed at room temperature for 1 h, including coating with rNP. For each of the three different assay incubation conditions, four individual sera of each species, each representing different levels of anti-RVF IgG antibody, were tested in quadruplicate.

### 2.8. IgG-Sandwich RVFV ELISA

To compare results between ELISAs based on a recombinant and based on a whole RVFV antigen, a subset of 27 sheep sera from the USA that tested negative by VNT but positive by IgG rNP I-ELISA when using the TG-ROC derived cut-off was assayed using an IgG-sandwich ELISA [38].

## 3. Results

### 3.1. VNT

Of the 9818 sera tested, 8818 were negative and 1000 were positive by VNT. All VNT-positive sera were from RVF-endemic countries. Irrespective of the geographic origin, all VNT-negative sera were regarded in this study as reference panels from RVFV non-infected animals, and VNT-positive sera were considered reference panels from animals infected with RVFV (Table 1).

### 3.2. Selection and Optimization of Cut-Off Values

Selection and optimization of cut-offs for the RVF IgG rNP I-ELISA in sheep, goats, and cattle using the TG-ROC and MCT analysis are shown in Figure 1. The optimization of cut-offs was based on the non-parametric program option [61,63] due to departure from a normal distribution of data analyzed (Figure 2). At cut-offs of 31.23 (A), 26.57 (B), and 30.46 (C), the overall MCT costs in sheep (A1), goats (B1), and cattle (C1) became minimal and were 0.0045, 0.0054, and 0.0625, respectively.

### 3.3. Distribution of rNP I-ELISA PP Values in Ruminant Sera from RVF-Endemic Countries and the Selection of Cut-Offs

The distribution of rNP I-ELISA PP values in VNT-negative and VNT-positive domestic ruminant sera from RVF-endemic countries and graphic illustration of the effect of different cut-offs on the test results are shown in Figure 3. Irrespective of the ruminant species tested and cut-off used, there was an overlap of PP I-ELISA values between VNT-positive and VNT-negative sera. The lowest number of false-positive results in VNT-negative sera was at TG-ROC selected cut-off and the highest at cut-off determined as mean + 2SD.

### 3.4. Distribution of rNP I-ELISA PP Values in Ruminant Sera from RVF-Free Countries and the Selection of Cut-Offs

The distribution of rNP I-ELISA PP values in VNT-negative ruminant sera from RVF-free countries and graphic illustration of the effect of different cut-off values on the I-ELISA results are shown in Figure 4. Irrespective of the ruminant species tested and different cut-offs used, the I-ELISA yielded some level of false-positive results. The lowest number of false-positive results was at cut-off determined as mean + 3SD and the highest at the cut-off determined as mean + 2SD.

### 3.5. D-Se and D-Sp

Estimates of D-Se at 95% and other measures of combined diagnostic accuracy of RVF rNP I-ELISA in ruminants from RVF-endemic countries are given in Table 2. The assay had the lowest estimates of D-Se in cattle, ranging from 90.7% to 95.8%, and was the lowest when TG-ROC and mean + 3SD derived cut-offs were applied. At all cut-offs, the D-Se in goats was 100%. The D-Se in sheep ranged from 97.5% to 99.6% and was the highest when mean + 2SD cut-off was applied.

The Ef estimates ranged from 0.947 to 0.967 and irrespective of the cut-off used were the lowest in cattle. Y estimates ranged from 0.893 to 0.988 and were the lowest in cattle at all cut-offs used. PPV estimates ranged from 83.8% to 97.12%, and all animals were the lowest when mean + 2SD cut-offs were applied. NPV estimates ranged from 97.2% to 100%, with the lowest in cattle and the highest in goats, irrespective of the cut-off used.

Estimates of D-Sp in subpopulations of ruminants from different RVF-endemic and RVF non-endemic countries are given in Table 3. In sheep from RVF-endemic countries, the D-Sp ranged from 92.7% to 96.6% at cut-off mean+2SD; from 97.6% to 98.4% at cut-off mean + 3SD; and from 98.6% to 100% at the TG-ROC-derived cut-off. Depending on the cut-off used, in sheep from RVF-non-endemic countries, the D-Se ranged from 96% to 98.9%. For the total of sheep serum panels from RVF-endemic countries, the D-Se ranged from 95.7% to 99.5%; for RVF non-endemic countries, it ranged from 96.5% to 97.7%; and for all sheep serum panels, it ranged from 96.7% to 98.6%, with the highest estimates of D-Sp recorded for all total panels when the TG-ROC derived cut-off was used (Table 3).

In goats from RVF-endemic countries, the D-Sp ranged from 94.7% to 95.7% at cut-off mean + 2SD, from 97.4% to 97.8% at cut-off mean + 3SD, and from 98.3% to 100% at TG-ROC derived cut-off. In a single panel of goat sera from RVF non-endemic countries, the D-Se ranged from 98.4% to 99.1%. For the total of goat serum panels from RVF-endemic countries, the D-Se ranged from 95.2% to 98.8%, and for all goat serum panels, it ranged from 96.5% to 98.9%, with the highest estimates of D-Sp recorded for all total panels when the TG-ROC derived cut-off was used (Table 3).

In cattle from RVF-endemic countries, the D-Sp ranged from 95.0% to 96.8% at cut-off mean + 2SD, from 96.8% to 98.86% at cut-off mean + 3SD, and from 93.6% to 100% at TG-ROC derived cut-off. Depending on the cut-off used, in sheep from RVF non-endemic countries, the D-Se ranged from 94.2% to 100%. For the total of sheep serum panels from RVF-endemic countries, the D-Se ranged from 96.3% to 98.7%; from RVF non-endemic countries, it ranged from 96.7% to 98.6%; and for all sheep serum panels, it ranged from 96.6% to 98.7%, with the highest estimates of D-Sp recorded for all total panels when the mean + 3SD cut-off was applied (Table 3).

The Mann–Whitney U-test for statistical analysis of the distribution of I-ELISA PP values showed that there were highly significant differences (*p* < 0.001) between PP values of subpopulations from the RVF-endemic and RVF-free groups per animal species of sheep, goats, or cattle (Table 4). The descriptive intercountry RVF-endemic pairwise comparison showed that all countries were significantly different except for Mozambique and Burkina Faso, which had similar PP means for each species of sheep, goats, or cattle. In addition, Yemen and Burkina Faso or Mozambique had similar PP means for sheep or cattle and Yemen and Uganda for goats.

### 3.6. The Effect of Inactivation

The different physicochemical inactivation methods used and evaluated in this study did not have an adverse effect on the kinetics and the detectable levels of the anti-RVFV IgG in positive ruminant sera. The titers (Table 5) as well as the dynamics of the dose-response curves (Figure 5) were similar before and after inactivation.

### 3.7. The Effect of Temperature Incubation

The PP values for high positive sera (S1-2, G1-2, and C1-2), mid-positive (S3, G3, and C3), and negative sera (S4, G4, and C4) were similar but slightly lower when all ELISA incubations were done at 37 °C for 1 h. Nonetheless, irrespective of the incubation method used, the assay was highly repeatable as evidence by low coefficient of variance (CV) values and with the use of different cut-offs determined in this study; all tested sera are correctly classified as positive and negative (Table 6).

### 3.8. IgG-Sandwich RVFV ELISA

Of 27 North American sheep sera that yielded negative results in VNT but positive results in IgG rNP I- ELISA when using TG-ROC derived cut-off (Table 3), 25 (92.6%) tested negative and two (7.4%) positive by IgG-sandwich RVFV ELISA.

## 4. Discussion

The continuous threat of re-emergence of large RVF outbreaks with dramatic veterinary and public health and socio-economic impacts in regions of endemicity and the risk of RVFV spread to RVF-free areas requires the development and application of safe, robust, high-throughput, and accurate diagnostic tests [71,72].

Diagnostic laboratories are tasked with providing analytical results that are based on internationally recognized methods and standards [73,74]. Assay validation is a part of quality assurance aiming at safeguarding qualification and competency and consequently indispensable in achieving accreditation status by an analytical laboratory [75]. A validated serological assay consistently provides test results that identify animals as seronegative or seropositive and by inference predicts the infection status of animals with a predetermined level of statistical certainty [62]. The epidemiology of RVF in endemic areas is still poorly understood and there is a need to evaluate low-cost surveillance tools, particularly for low- and middle-income settings (LMICs) [60].

Traditional antigen production methods pose health risks, thus restricting their use to high biosafety level facilities. Furthermore, due to the poor binding of these antigens to ELISA plates, it is necessary to use more complex and time-consuming ELISA formats, including sandwich, capture, or inhibition ELISA techniques [38,40]. In addition, the binding of antibodies to cellular contaminants present in RVFV whole-antigen preparations may lead to cross-reactivity, resulting in reduced specificity. To overcome these limitations, high-quality recombinant antigens can be safely produced outside high-bio-containment facilities.

An I-ELISA represents the most simple, cost-effective, and easy-to-automate immunoassay [76]. The RVFV NP is a supreme antigen for I-ELISA due to its high immunogenicity [19,20] and high nucleic acid conservation [14,77]. Antigenic cross-reactivity studies using sera from experimentally infected sheep demonstrated that antibodies induced by African phleboviruses other than RVFV should not confuse serodiagnosis of RVF [78]. RVFV rNP-based I-ELISA has been reported to not cross-react with sera from mice experimentally infected with different viruses of genus *Phlebovirus*, *Nairovirus* and *Orthobunyavirus* [57]. Other advantages of RVFV rNP-based I-ELISA, including cost-effectiveness of rNP antigen production, have been previously discussed [43,44].

Ideally, the diagnostic performance of a serological assay should be determined by testing sera from subjects of known infection status. The diagnostic threshold or cut-off represents an assay value used to dichotomize negative and positive results and, by inference, to define the infection status of an individual. The relevance of data used for the determination of cut-off consequently affects estimates of test performance. Gold standards for the selection of truly infected and uninfected subjects include isolation of the agent or pathognomonic histopathological criteria. In practice, a true gold standard is difficult to accomplish; thus, relative standards of comparison are used [62]. In this study, we used the VNT test to define the RVFV infection status of animals tested. Although virus neutralization assays are generally considered reference tests and are still used in reference diagnostic laboratories, they are labor-intensive, time-consuming, and require expensive bio-containment facilities [14]; thus, they are not easily affordable and impractical for surveillance activities, particularly in LMICs [60].

Various statistical analyses used in our study for the selection of the cut-off values provided similar results. A cut-off value determined as two or three SD above the mean in uninfected individuals is frequently used for the interpretation of serodiagnostic tests. However, this approach assumes a normal distribution of the test values in the population targeted by an assay and provides only an estimate of D-Sp [62]. Deviations from normality are often recorded in serological data and should be addressed in the selection of diagnostic threshold values [79]. Therefore, we also used the TG-ROC analysis for the selection and optimization of cut-off values to account for parametric versus nonparametric distribution of test values. Due to the inherent differences amongst assay systems, binding-antibody levels should be expressed in relative rather than absolute terms. One of the advantages of conversion OD readings into PP values of the positive serum standard as a measure of antibody activity in the I-ELISA is that this method does not assume a uniform background activity, and therefore it is more appropriate for inter-laboratory standardization [61].

We analyzed and compared the diagnostic performance of RVFV rNP-based I-ELISA in geographically separate populations and different domestic ruminant species in both RVF-endemic and RVF-free countries. The results demonstrate that the test performs consistently across different animal groups investigated with high estimates of D-Sp and D-Se and other measures of diagnostic accuracy. The observed variations across the different subpopulations tested may be due to the relatively small numbers of animals representing each specific subpopulation. To account for the distribution of covariate factors (age, sex, genetic, nutritional, geographical, and stage of infection) that may influence the estimates of an ELISA diagnostic accuracy, the targeted populations of animals should preferably be sampled using simple random, systematic, or stratified sampling methods [64]. These ideal conditions are difficult to achieve in practice and could not be applied during this study. Sera used in this study represent referral diagnostic submissions, and except for their animal and geographic origins and categorization by the VNT results, no other covariate factors that might influence the estimates of performance characteristics of the RVFV rNP-based I-ELISA were analyzed. While the influence of referral submissions on the performance of a diagnostic test under validation should be considered, this seems to not apply to collections of sera from laboratories that are involved in large-scale routine testing [63]. Time-dependent changes in the sensitivity of an assay may be of significance for epidemic situations, where the stage of disease may affect the outcome of assay results and interpretation of data. However, in practice, the impacts and interrelationship of multiple factors are mostly unknown, and assay diagnostic accuracy estimates are based on average values calculated in non-homogeneous populations. Biological variables possibly contribute more significantly to false-positive than to false-negative results [62].

Therefore, to account for the probable increased variance that would affect the estimates of the diagnostic specificity, relatively large numbers of animals from RVF-free countries were analyzed in the present study. From the point of using the VNT in classifying animals as infected or non-infected, it should be noted that infection with RVFV induces rapid appearance and long-lasting IgG antibodies following RVFV infection [9,14], making their detection a useful tool for diagnosis of outbreaks, epidemiological investigations, and disease risk studies. There is also no evidence of serological subgroups or major antigenic variations between RVFV isolates of disparate chronologic or geographic origins [6]. In this context, the use of the serological gold standard in dichotomizing animals according to their infection status and the relatively large total numbers of animals from RVF-endemic and RVF-free areas seems to ensure statistically reliable estimates of rNP-based I-ELISA diagnostic performance.

In this study, the TG-ROC MCT-optimized cut-offs were selected under the assumption that the cost of false-positive and false-negative results are equal and the prevalence of infection was 50%. The prevalence assumed in our study may not be the same in other populations targeted by the assay, and this should be noted when applying the estimates of diagnostic accuracy reported for the RVFV rNP-based I-ELISA in the present work.

The diagnostic accuracy estimates determined in our study are similar to those previously published for ELISAs based on a whole RVFV antigen [36,37,38,39,40,41,55] as well as those based on recombinant NP ELISAs [51,52,53,54,55,56,57,58,59,60], including the commercially available NP-based competition ELISAs [52,60]. Although recorded at a relatively low rate, false-positive results were documented in this study across different subpopulations and are of particular concern in ruminants originating from RVF-non-endemic countries. A similar problem has been reported in North American sheep [46]. The reasons for the anti-RVFV NP cross-reacting IgG antibodies remain unknown, but it has been postulated that potential cross-reactivity might be caused by unidentified closely related agent(s) circulating among ruminants in RVF-free areas or potential cross-reaction with antibodies raised against commensal or pathogenic *E. coli* in ruminants [46]. Both assumptions seem to be supported by the results of our study. While most of VNT-negative sheep sera from North America that tested positive by rNP-based I-ELISA were negative by IgG-sandwich RVFV ELISA, 7.4% tested positive by the latter assay, which is based on a whole antigen of RVFV [38]. The lower D-Se of rNP-based I-ELISA in cattle recorded in our study was also reported for a commercial competitive ELISA based on recombinant NP in Cameroonian cattle with D-Se ranging from 84.4% to 98.1% between different subpopulations tested [60]. The reason for lower D-Se remains unknown but is likely due to the higher cross-reactiveness of cattle sera.

Unlike *E. coli*, baculovirus seems to be not infectious for ruminant domestic livestock, which makes it an appropriate virus for the production of recombinant animal diagnostic antigens. Promising preliminary performance of the RVFV I-ELISA based on the use of the baculovirus-expressed recombinant NP antigen was recently reported in sera from RVFV experimentally infected sheep and calves and in sera from indigenous sheep and goats naturally infected in the Gambia [46].

Inactivation protocols evaluated in our study indicate that they do not affect detectable levels of anti-RVFV-IgG. Likewise, different incubation procedures evaluated in this study did not have an adverse effect on the detectable levels of the anti-RVFV-IgG in ruminant sera, repeatability, or interpretation of the test results.

Achieving the required level of assay validation is impossible without international collaboration and willingness to share clinical materials between countries both from disease-endemic and non-endemic regions. This work constitutes the largest validation study undertaken for rNP-based I-ELISA to date and significantly contributes to research-collaborating activities aiming at improving surveillance tools for zoonotic pathogens of global concern and their application in less-resourced laboratories.

In summary, the performance characteristics of RVFV rNP-based I-ELISA determined in this study further demonstrate that the assay is a valuable surveillance tool for the detection of IgG anti-RVFV in domestic ruminants. The relatively low-cost and easy-to-perform I-ELISA format makes it a suitable diagnostic test for LMICs. The standard incubation and inactivation procedures evaluated did not have an adverse effect on the detectable levels of the anti-RVFV IgG in ruminant sera and thus together with recombinant antigen-based I-ELISA provide a simple, safe, and robust diagnostic platform.

## Figures and Tables

**Figure 1 viruses-13-01651-f001:**
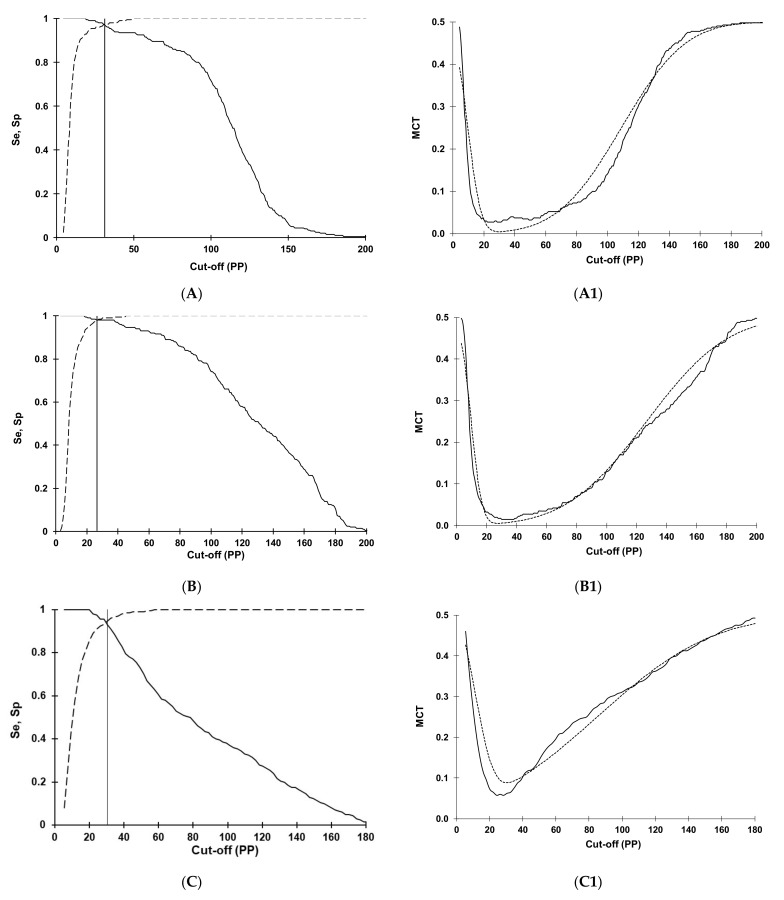
Optimization of cut-offs for the Rift Valley fever IgG indirect ELISA based on recombinant nucleocapsid antigen in sheep (**A**), goats (**B**), and cattle (**C**) using the two-graph receiver operating characteristic analysis (TG-ROC). The insertion point of the sensitivity (Se, smooth line) and specificity (Sp, dashed line) graphs represents a cut-off PP value (31.23, 26.57, and 30.46, respectively) at which the highest and equivalent test parameters (Se = Sp) are achieved at 95% accuracy level. Using the misclassification cost term (MCT) option of the TG-ROC, at these cut-off values, the overall misclassification costs in sheep (**A1**), goats (**B1**), and cattle (**C1**) become minimal (0.0045, 0.0054, and 0.0625, respectively) under the assumption of 50% disease prevalence and equal costs of false-positive and false-negative results. The two MCT curves represent values based on non-parametric (smooth line) or parametric (dashed line) estimates of Se and Sp derived from datasets in field-collected sera.

**Figure 2 viruses-13-01651-f002:**
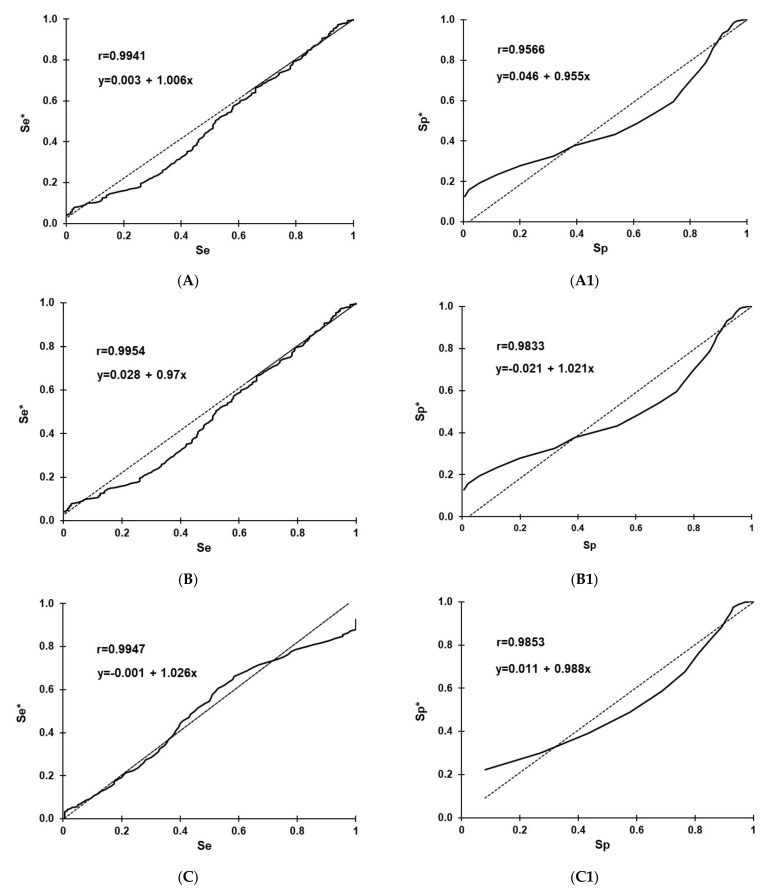
Linear correlation analysis of the non-parametric (smooth line) sensitivity (Se) vs. parametric (dashed line) sensitivity (Se*) in sheep (**A**), goats (**B**), and cattle (**C**), and the non-parametric specificity (Sp) vs. parametric specificity (Sp*) in sheep (**A1**), goats (**B1**) and cattle (**C1**).

**Figure 3 viruses-13-01651-f003:**
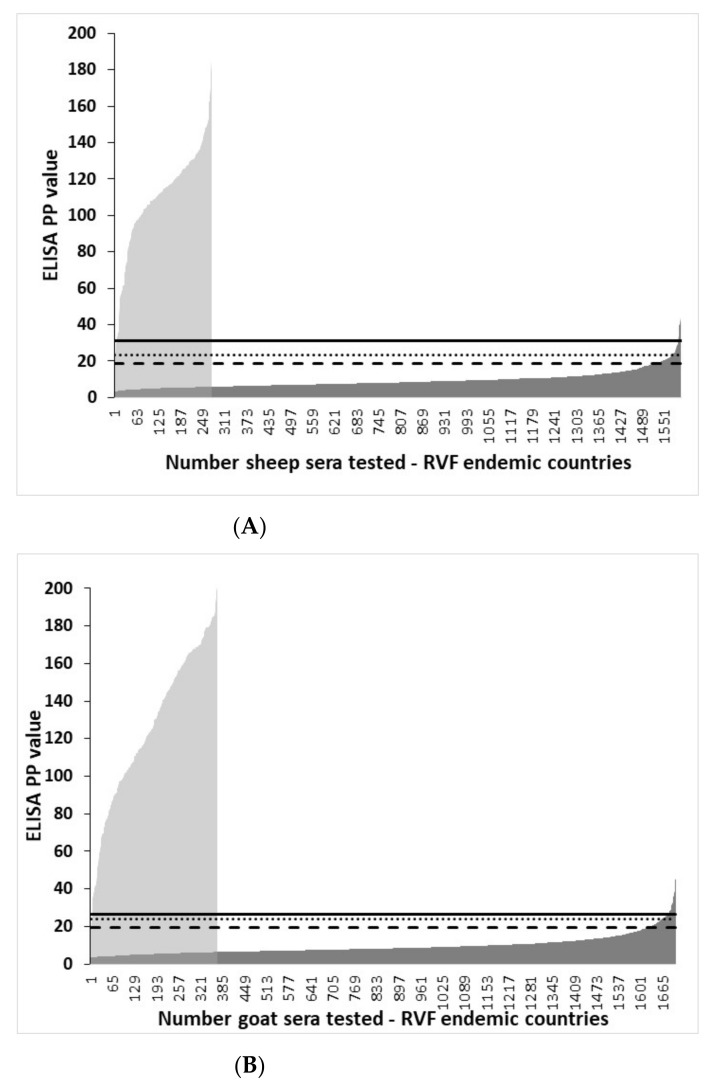
The effect of different I-ELISA cut-off values on the categorization between sera from RVF-endemic countries tested positive or negative in the virus neutralization test. Distribution of IgG I-ELISA PP values in sera tested positive in VNT (grey area): (**A**) sheep (*n* = 275), (**B**) goats (*n* = 369), and (**C**) cattle (*n* = 356). Distribution of IgG I-ELISA PP values in sera tested negative in VNT (dark area): (**A**) sheep (*n* = 1874), (**B**) goats (*n* = 2072), and (**C**) cattle (*n* = 2864) in the VNT. Sera ordered according to ELISA PP values. Vertical lines indicate the ELISA cut-off values determined by the TG-ROC analysis (solid line), and as mean plus three (dotted line) and two (slashed line) standard deviations observed in the VNT-negative sera.

**Figure 4 viruses-13-01651-f004:**
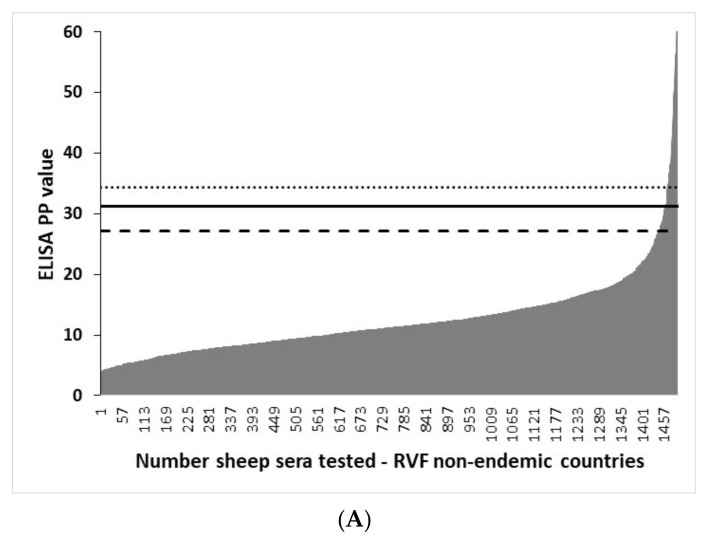
The effect of different I-ELISA cut-off values on the categorization of sera from RVF-non-endemic countries tested negative in the virus neutralization test. Distribution of IgG I-ELISA PP values (grey area) in (**A**) sheep (*n* = 1493), (**B**) goats (*n* = 560), and (**C**) cattle (*n* = 955). Sera ordered according to ELISA PP values. Vertical lines indicate the ELISA cut-off values determined by the TG-ROC analysis (solid line) in ruminant serum panels from RVF-endemic countries and as mean plus three (dotted line) and as mean plus two (slashed line) standard deviations observed in the VNT-negative serum panels from RVF-free countries.

**Figure 5 viruses-13-01651-f005:**
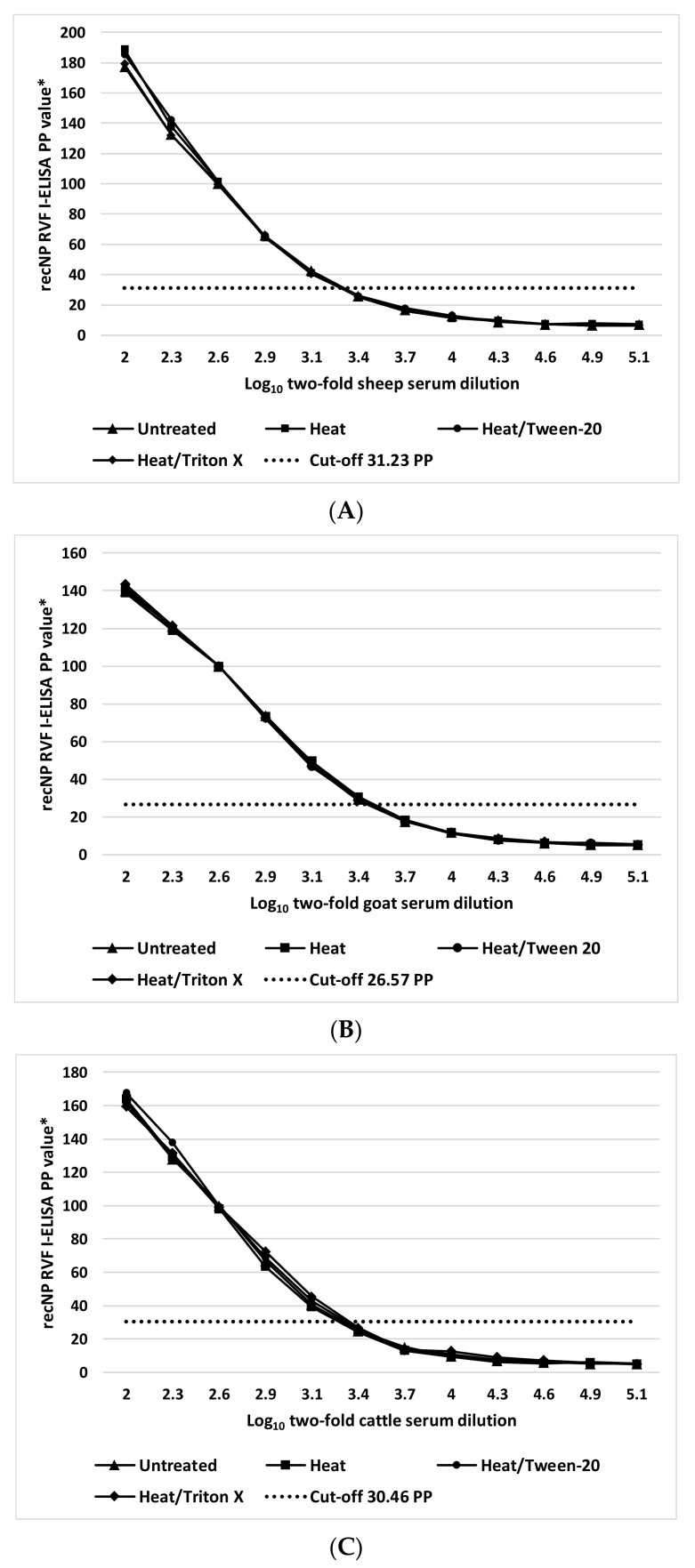
Dose-response kinetics of a positive RVFV IgG sheep (**A**), goat (**B**), and cattle (**C**) serum before and after different inactivation procedures measured by recombinant nucleocapsid I-ELISA. * Percent positivity of internal positive control serum.

**Table 1 viruses-13-01651-t001:** Origin and number of domestic ruminant sera from Rift Valley fever endemic and RVR free countries and the results of virus neutralization test.

Country	Ovine		Total	Caprine		Total	Bovine		Total
	VNT^+ 1^	VNT^− 2^	Tested	VNT^+^	VNT^−^	Tested	VNT^+^	VNT^−^	Tested
RVF-endemic									
Burkina Faso	20	165	185	0	268	268	72	924	996
DRC	1	61	62	0	0	0	133	813	946
Mozambique	170	435	605	171	717	888	0	0	0
Senegal	16	234	250	2	76	78	58	94	152
Uganda	0	0	0	170	527	697	60	623	683
Yemen	68	704	772	26	115	141	33	54	87
Sub-total	275	1599	1874	369	1703	2072	356	2508	2864
RVF-free									
France	0	720	720	0	560	560	0	640	640
Poland	0	0	0	0	0	0	0	292	292
USA	0	773	773	0	0	0	0	23	23
Sub-total	0	1493	1493	0	560	560	0	955	955
Total	275	3092	3367	369	2263	2632	356	3463	3819

^1^ Number of sera tested positive in virus neutralization test; ^2^ Number of sera tested negative in virus neutralization test.

**Table 2 viruses-13-01651-t002:** Diagnostic sensitivity and other accuracy measures of rNP I-ELISA for the detection of anti-IgG RVFV antibody in domestic ruminants from RVF-endemic countries using different cut-off values.

Animal Species	Cut-Off PP Value ^1^	FN ^2^/TP ^3^	D-Se ^4^ (95% CI) ^5^	Ef ^6^	Y ^7^	PPV ^8^ (%)	NPV ^9^ (%)
Sheep							
Mean + 2SD ^10^	18.63	1/274	99.6 (98–100)	0.964	0.953	83.80	99.91
Mean + 3SD ^11^	23.30	2/273	99.3 (97.4–99.9)	0.963	0.976	91.75	99.86
TG-ROC ^12^	31.23	7/268	97.5 (94.8–99)	0.960	0.970	97.12	99.57
Goats							
Mean + 2SD	19.28	0/369	100 (99–100)	0.967	0.952	86.82	100.00
Mean + 3SD	24.17	0/369	100 (99–100)	0.966	0.978	92.62	100.00
TG-ROC	26.57	0/369	100 (99–100)	0.966	0.988	95.63	100.00
Cattle							
Mean + 2SD	24.38	15/341	95.8 (93.1–97.6)	0.957	0.921	88.65	98.70
Mean + 3SD	30.86	33/323	90.7 (87.2–93.5)	0.948	0.894	94.22	97.85
TG-ROC	30.46	33/323	90.7 (87.2–93.5)	0.947	0.893	93.86	97.82

^1^ Percentage positivity of the positive internal control serum; ^2^ False negative; ^3^ True positive; ^4^ Diagnostic sensitivity; ^5^ Confidence intervals; ^6^ Efficiency; ^7^ Youden’s index; ^8^ Positive predictive value;^9^ Negative predictive value; ^10^ Cut-off value determined as mean + 2 standard deviations of PP values in VNT-negative animals; ^11^ Cut-off value determined as mean + 3 standard deviations of PP values in VNT-negative animals; ^12^ Cut-off value determined by the two-graph receiver operating characteristic analysis in ruminant serum panels from RVF-endemic countries.

**Table 3 viruses-13-01651-t003:** Diagnostic specificity of rNP I-ELISA for the detection of anti-IgG RVFV antibody in domestic ruminants from RVF-endemic countries.

Species/Origin	No. VNT ^1^	Cut-Off PP ^2^	FP ^3^/TN ^4^	Mean + 2SD ^5^ D-Sp ^6^ (95%CI) ^7^	Cut-Off PP	FP/TN	Mean + 3SD ^8^ D-Sp (95% CI)	Cut-Off PP	FP/TN	TG-ROC ^9^ D-Sp (95% CI)
Sheep serum panels										
Burkina Faso	165	16.8	12/153	92.7 (87.6–96.2)	20.6	4/161	97.6 (93.9–99.3)	31.23	0/165	100 (97.8–100)
DRC	61	11.29	2/59	96.7 (88.7–99.6)	13.63	1/60	98.4 (91.2–100)	31.23	0/61	100 (94.1–100)
Mozambique	435	20.74	16/419	96.3 (94.1–97.9)	26.54	9/426	97.9 (96.1–99)	31.23	6/429	98.6 (97–99.5)
Senegal	234	21.15	8/226	96.6 (93.4–98.5)	26.17	4/230	98.3 (95.7–99.5	31.23	1/233	96.6 (97.6–100)
Yemen	704	16.72	40/664	94.3 (92.3–95.9)	20.57	17/687	97.6 (96.2–98.6)	31.23	1/703	99.9 (99.2–100)
France	720	24.01	17/703	97.6	29.29	1o/710	98.6	31.23	8/712	98.9
USA	773	29.19	31/742	96.0	37.84	19/754	97.5	31.23	27/746	96.5
Total RVF-endemic panels	1599	18.63	69/1530	95.7 (94.6–96.6)	23.30	27/1572	98.3 (97.6–98.9)	31.23	8/1591	99.5 (99–99.8)
Total RVF-free panels	1493	27.17	52/1441	96.5	34.43	29/1464	98.1	31.23	35/1458	97.7
Total all panels	3092	23.48	103/2989	96.7 (96–97.3)	29.77	49/3043	98.4 (97.9–98.8	31.23	43/3049	98.6 (98.1–99)
Goat serum panels										
Burkina Faso	268	17.37	13/255	95.2	21.79	6/262	97.8	26.57	0/268	100 (98.8–100)
Mozambique	717	17.81	37/680	94.8 (93–96.3)	22.3	16/701	97.8 (96.4–98.7)	26.57	7/710	99 (98–99.6)
Senegal	76	27.18	4/72	94.7 (87.1–98.5)	34.16	2/74	97.4 (90.8–99.7)	26.57	4/72	94.7 (87.1–98.5)
Uganda	527	20.02	28/499	94.7 (92.4–96.4)	24.93	12/515	97.7 (96.1–98.8)	26.57	8/519	98.5 (97–99.3)
Yemen	115	20.03	5/110	95.7 (90.1–98.6)	24.75	3/112	97.4 (92.6–99.5)	26.57	2/113	98.3 (93.9–99.8)
France	560	22.66	9/551	98.4	27.91	5/555	99.1	26.57	5/555	99.1
Total RVF-endemic panels	1703	19.28	82/1621	95.2 (94.1–96.2)	24.17	37/1666	97.8 (97–98.5)	26.57	21/1682	98.8 (98.1–99.2)
Total RVF-free panels	560	22.66	9/551	98.4	27.91	5/555	99.1	26.57	5/555	98.1
Total all panels	2263	20.39	80/2183	96.5 (95.6–97.2)	25.49	35/2228	98.5 (97.9–98.9)	26.57	26/2237	98.9 (98.3–99.2)
Cattle serum panels										
Burkina Faso	924	25.23	35/889	96.2 (94.8–97.3)	32.38	16/908	98.3 (97.2–99)	30.46	18/906	98.1 (96.9–98.8)
DRC	813	22.58	32/781	96.1 (94.5–97.3)	28.45	9/804	98.9 (97.9–99.5)	30.46	7/806	99.1 (98.2–99.7)
Senegal	94	33.39	3/91	96.8 (91–99.3)	41.56	3/91	96.8 (91–99.3)	30.46	6/88	93.6 (86.6–97.6)
Uganda	623	22.96	31/592	95 (93–96.6)	28.46	9/614	98.6 (97.3–99.3)	30.46	4/619	99.4 (98.4–99.8)
Yemen	54	22.74	2/52	96.3 (87.3–99.5)	28.14	1/53	98.1 (90.1–100)	30.46	0/54	100 (93.4–100)
France	640	23.35	12/628	98.1	27.68	5/635	99.2	30.46	3/637	99.5
Poland	292	34.44	12/280	95.9	43.41	11/281	96.2	30.46	17/275	94.2
USA	23	15.42	1/22	95.7	17.47	0/23	100	30.46	0/23	100
Total RVF-endemic panels	2508	24.38	94/2414	96.3 (95.4–97)	30.86	32/2476	98.7 (98.2–99.1)	30.46	35/2473	98.6 (98.1–99)
Total RVF-free panels	955	27.53	32/923	96.7	33.71	13/942	98.6	30.46	20/935	97.9
Total all panels	3463	25.68	118/3345	96.6 (95.9–97.2)	32.30	45/3418	98.7 (98.3–99.1)	30.46	55/3408	98.4 (97.9–98.8)

^1^ Number of sera tested negative in virus neutralization test; ^2^ Percentage of the positive internal control serum used in recNP I-ELISA; ^3^ False positive; ^4^ True negative; ^5^ Cut-off value determined as mean + 2 standard deviations of PP values in VNT-negative animals; ^6^ Diagnostic specificity; ^7^ Confidence intervals; ^8^ Cut-off value determined as mean + 3 standard deviations of PP values in VNT-negative animals; ^9^ ELISA cut-off value determined by the two-graph receiver operating characteristic analysis in ruminant serum panels from RVF-endemic countries.

**Table 4 viruses-13-01651-t004:** Statistical analysis of the distribution of rNP I-ELISA PP values in RVFV-free subpopulations from RVF-endemic and non-endemic countries.

Species/Origin	Shapiro-Wilk Test ^1^	Dunn’s Test ^2^	Mean/Median PP ^3^	SD ^4^/(IQR) ^5^ PP	Range PP
VNT-negative sheep sera					
RVF-endemic countries	<0.001	<0.001	9.3/8.2	4.7/(6.4–10.6)	1.8–49.2
RVF-free countries	<0.001		12.6/11.3	7.3/(8.4–14.7)	3.9–75.8
VNT-negative goat sera					
RVF-endemic countries	<0.001	<0.001	9.5/8.2	4.9/(6.6–10.8)	2.3–45.2
RVF-free countries	<0.001		12.2/11.8	5.2/(8.5–14.8)	4.4–69.2
VNT-negative bovine sera					
RVF-endemic countries	<0.001	<0.001	11.4/9.7	6.5/(7.1–13.8)	3.3–76.0
RVF-free countries	<0.001		15.2/14.5	6.2/(11.5–17.8)	4.9–68.0

^1^ Shapiro–Wilk test for normality; ^2^ Mann–Whitney U-test Dunn’s non-parametric analysis of variance to assess difference between means of I-ELISA PP values in subpopulations tested; ^3^ ELISA percent positivity; ^4^ Standard deviation; ^5^ Interquartile range.

**Table 5 viruses-13-01651-t005:** Comparison of rNP I-ELISA ELISA mean titers and analytical slopes of non-treated versus inactivated IgG RVFV positive sheep, goat, and cattle sera.

Assay (Cut-Off PP Value) ^1^	Mean log_10_ Serum Titre ^2^/Dose Response Curve R Square ^3^			
	Untreated	60° 1 h	0.5%Tween 20 15 min 60°	0.5%Triton X-100 15 min 60°
Ovine (31.23)	3.1/0.9665	3.5/0.9603	3.5/0.9625	3.5/0.9651
Caprine (26.57)	3.1/0.9572	3.5/0.9584	3.5/0.9590	3.5/0.9603
Bovine (30.46)	3.1/0.9638	3.2/0.9607	3.2/0.9625	3.2/0.9640

^1^ TG-ROC derived cut-off; ^2^ Log_10_ highest serum dilution at which at least 75% of four replicates tested positive; ^3^ Coefficient of determination.

**Table 6 viruses-13-01651-t006:** The effect of different incubation temperatures on recombinant nucleocapsid RVFV indirect ELISA results in domestic ruminant sera.

Incubation Conditions	4 °C/37 °C ^1^			37°C/37 °C ^2^			RT/RT ^3^		
Serum ^4^	Mean PP ^5^	SD ^6^	CV ^7^	Mean PP	SD	CV	Mean PP	SD	CV (%)
S1	130.96	2.09	1.60	125.09	1.23	1.01	132.51	1.63	1.11
S2	122.95	3.02	2.45	121.25	2.65	2.18	123.80	1.12	0.91
S3	62.70	2.63	4.20	59.64	1.54	2.76	65.65	1.30	1.98
S4	13.50	0.85	5.47	10.26	0.45	4.43	12.57	0.76	6.03
G1	180.70	2.13	1.18	175.94	0.41	0.25	180.08	1.37	0.76
G2	117.35	0.91	0.77	108.77	1.09	1.00	111.50	2.66	2.38
G3	73.11	1.79	2.44	73.89	1.13	1.53	70.60	1.58	2.24
G4	12.46	0.73	5.86	8.83	0.23	2.60	10.62	0.30	2.79
C1	132.56	2.72	2.05	126.96	2.63	1.92	135.68	1.63	1.19
C2	117.29	0.76	0.64	116.37	3.13	2.64	120.48	1.10	0.92
C3	44.06	0.46	1.04	44.02	2.01	4.56	43.52	2.12	4.87
C4	15.27	0.55	3.63	12.87	0.33	3.03	15.40	0.51	3.32

^1^ ELISA plate coated with rNP overnight at 4 °C and all subsequent incubations (except for substrate addition) performed at 37 °C for 1 h; ^2^ ELISA plate coated with rNP and all subsequent incubations (except for substrate addition) performed at 37 °C for 1 h; ^3^ ELISA plate coating with rNP and all subsequent incubations performed at room temperature for 1 h; ^4^ Each of four sheep (S1-4), goat (G1-4), and cattle (C1-4) sera were tested in quadruplicate; ^5^ mean percentage positivity (PP) value of four replicates; ^6^ standard deviation; ^7^ coefficient of variation.

## Data Availability

The data presented in this study are available on request from the corresponding author.

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
