# Peer review of "Large-Scale International Validation of an Indirect ELISA Based on Recombinant Nucleocapsid Protein of Rift Valley Fever Virus for the Detection of IgG Antibody in Domestic Ruminants"

_viruses, 2021, doi:10.3390/v13081651_

Round 1

Reviewer 1 Report

The manuscript describes a validation for an ELISA system (J. Virol. Methods 2007) for serological determination of Rift Valley fever virus (RVFV) infection using serum from ruminants collected in RVFV non-endemic and endemic areas.  This work is not particularly original because similar serological approaches have been described by this group and other groups. In addition, ELISA kits for serological detection of RVFV are commercially available from different vendors. In conclusion, the main concern for this work is the lack of originality. 

I have the following minor suggestions for such work:

1) Page 2 (Introduction). Please include the following information: RVFV is suspected to induce miscarriages in women. Rift Valley Fever: a Threat to Pregnant Women Hiding in Plain Sight? Cynthia M McMillen, Amy L Hartman 2021. 

2) Page 4 (Virus Neutralization Test). Which virus has been used for the neutralization assay? Please include information about the viral strain and the presence of a possible marker like GFP. 

3) Page 4 (ELISA). Please include the sequence information for NP (e.g. GenBank and virus strain).

4) Figure 1 and 2. Please use bigger fonts inside the graphs for better clarity.

5)  Table 3. Please organize the table and improve column alignment.

6) Table 5 is not visible.

7) Page 7 (Discussion). "Unlike E. coli, baculovirus seems to be not infectious for ruminant domestic livestock, which makes it an appropriate virus for the production of recombinant animal diagnostic antigens." How E. coli used for the production of the antigen would infect the livestock?   

8) Discussion seems very long, I would recommend improving its conciseness. 

9) Please include line numbers according to the journal guidelines.

Author Response

Reviewer 1 - We thank for valuable comments.

Comments and Suggestions for Authors

The manuscript describes a validation for an ELISA system (J. Virol. Methods 2007) for serological determination of Rift Valley fever virus (RVFV) infection using serum from ruminants collected in RVFV non-endemic and endemic areas.  This work is not particularly original because similar serological approaches have been described by this group and other groups. In addition, ELISA kits for serological detection of RVFV are commercially available from different vendors. In conclusion, the main concern for this work is the lack of originality. 

Response: Please note that availability of commercial kits for serodiagnosis of RVF is very limited, they are extremely expensive and hardly affordable by developing countries, and particularly by RVF endemic African countries. None of the commercial RVF ELISA kit is produced in Africa and most veterinary diagnostic labs are dependent on donation of these kits by UN international organizations.

The ELISA validated in our study is based on detailed published information how it was originally developed. Production of recombinant RVFV nucleocapsid antigen is cost-efficient; it takes two days to produce the ELISA recombinant antigen for testing more than 30 000 animal sera in duplicate.

Other advantages of the ELISA we validated are discussed in the submitted manuscript. Our work constitute an important part of operational research, much need in the developing countries and it represents the largest RVF ELISA validation study in veterinary science based on international collaboration – an important and very encouraging example of what can be achieved when we willing to collaborate and share research/diagnostic materials. Progress in nowadays science is not possible without networking.

Our work might be not “original”, but science/research does not have to be original to be important for veterinary and public health. Many types of covid-19 vaccines were/are developed using already well-known and established laboratory technology (not original), yet very important to combat the ongoing pandemic.

Our work is highly appreciated by veterinary laboratory diagnosticians, and we believe that publication of this manuscript will encourage in-house production of diagnostic reagents to avoid spending a lot of money for commercial assays from limited sources. For any interested laboratory we can supply free of charge plasmids for the production of recNP I-ELISA antigen.

I have the following minor suggestions for such work:

1) Page 2 (Introduction). Please include the following information: RVFV is suspected to induce miscarriages in women. Rift Valley Fever: a Threat to Pregnant Women Hiding in Plain Sight? Cynthia M McMillen, Amy L Hartman 2021. 

Response: Information on suspected induction of miscarriages in women and the relevant reference included as suggested and the following reference added: McMillen, C.M.; Hartman, A.L. 2021. Rift Valley fever: a threat to pregnant women hiding in plain sight? J. Virol. 2021, 95:e01394-19.

2) Page 4 (Virus Neutralization Test). Which virus has been used for the neutralization assay? Please include information about the viral strain and the presence of a possible marker like GFP. 

Response: Information is provided as requested. “The virus neutralization test was performed using the AR 20368 strain of RVFV isolated in 1981 from Culex zombaensis in South Africa”, as per provided reference 37 in original submission (in revised manuscript reference 38) (Paweska et al., 2003). This is a wild type isolate. VNT test readout relying on cytopathic effects, and therefore does not include a fluorescent marker like GFP.

3) Page 4 (ELISA). Please include the sequence information for NP (e.g. GenBank and virus strain).

Response: Sequence information included as requested “The NP sequence is based on the Zim688/78 RVFV strain isolated from a bovine in 1978 in Zimbabwe. The coding sequence is available in Genbank, accession number DQ924959”. This information is was available in reference 43 in original submission (in revised manuscript reference 44) (Jansen van Vuren et al., 2007).

4) Figure 1 and 2. Please use bigger fonts inside the graphs for better clarity.

Response: Changed as suggested. Also please note that Fig. B should be Fig. A, Fig. A1 should be B1 – corrected accordingly in the revised manuscript.

5)  Table 3. Please organize the table and improve column alignment.

Response: Done as recommended.

6) Table 5 is not visible.

Response: We apologize for this, but it was uploaded on Viruses submission portal with original manuscript. Likely disappeared while our original submission was converted to Viruses’ manuscript format. Included in the revised manuscript on version provided by Editorial Office of Viruses. Unfortunately the page on which the table should appear is not easy editable and once word format of the table is inserted it disappears again, and typing in on this page is difficult. For this reason a pdf copy of the table is inserted.

7) Page 7 (Discussion). "Unlike E. coli, baculovirus seems to be not infectious for ruminant domestic livestock, which makes it an appropriate virus for the production of recombinant animal diagnostic antigens." How E. coli used for the production of the antigen would infect the livestock?   

Response: Obviously it will not, but E. coli system used for expression of recombinant antigen likely has cross-reacting antigens with other E. coli strains, and despite purification steps used for production of recombinant antigen it might be that this preparation is not 100% pure - contaminated with E. coli antigen(s), and consequently causing ELISA unspecific background when tested sera from animals who were/are infected with E.coli. This is a working hypothesis as discussed in the paper.

8) Discussion seems very long, I would recommend improving its conciseness. 

Response: We shorten the discussion to make it more conciseness as suggested.

 9) Please include line numbers according to the journal guidelines.

Response: Included in the revised manuscript.

Reviewer 2 Report

Comments on Paweska et al., “Large-scale international validation of an indirect ELISA based on recombinant nucleocapsid protein of RVFV for the detection of IgG antibody in domestic ruminants”

This is a first-of-its-kind large scale validation study of the use of an indirect-ELISA employing nucleocapsid protein generated from recombinant DNA.  The study is very thorough and demonstrates that such an I-ELISA assay provides a robust, sensitive, selective diagnostic for the presence of IgG antibodies resulting from RVFV infection.  Importantly, in addition to exhibiting good sensitivity and selectivity under a variety of experimental conditions, generation of the ELISA itself is safer and more scaleable than using a comparable assay employing virally-derived antigens from infected cells.  Another aspect that makes this work significant is its large size and the considered use of samples from animals in both RVFV-endemic and non-endemic areas, which bolsters confidence in robustness of the assay.  The results of this study should have significant practical impacts on regulation and monitoring of global trade of animals and animal products.

I have several relatively minor comments about the content/layout of the manuscript.

1) Although the thorough Introduction is appreciated for context of the following sections, there is considerable redundancy in the content of the Intro and Discussion sections.

2) The Material and Methods section is pretty thorough, but some of the descriptions of the statistical/mathematical tools used to analyze the data are not really geared to a ‘virological’ audience.  If this work is to have more impact on this audience, it would be helpful to explain some of these tools in less jargony language.  For example, explaining what is a TG-ROC analysis and why do you use this could provide some context for people outside this area.

3) p5, section 2.6: “laboratory” is misspelled

4) p 11, section 3.5 (and/or in Discussion), it would be of interest to speculate on the relatively lower D-Se for cattle compared to goats, sheep.

5) Table 3 formatting is off and makes it very hard to read.

6) Table 5 appears to lack any content

7) Reference 34 in the bibliography should read “Wichgers Shreur, P.J….” instead of “Schreur, P.W.”

Author Response

Reviewer 2 - We thank  for valuable comments.

Comments and Suggestions for Authors

Comments on Paweska et al., “Large-scale international validation of an indirect ELISA based on recombinant nucleocapsid protein of RVFV for the detection of IgG antibody in domestic ruminants”

This is a first-of-its-kind large scale validation study of the use of an indirect-ELISA employing nucleocapsid protein generated from recombinant DNA.  The study is very thorough and demonstrates that such an I-ELISA assay provides a robust, sensitive, selective diagnostic for the presence of IgG antibodies resulting from RVFV infection.  Importantly, in addition to exhibiting good sensitivity and selectivity under a variety of experimental conditions, generation of the ELISA itself is safer and more scalable than using a comparable assay employing virally-derived antigens from infected cells.  Another aspect that makes this work significant is its large size and the considered use of samples from animals in both RVFV-endemic and non-endemic areas, which bolsters confidence in robustness of the assay.  The results of this study should have significant practical impacts on regulation and monitoring of global trade of animals and animal products.

I have several relatively minor comments about the content/layout of the manuscript.

1) Although the thorough Introduction is appreciated for context of the following sections, there is considerable redundancy in the content of the Intro and Discussion sections.

Response: We removed redundancy in the content of the Intro and Discussion sections as recommended.  

2) The Material and Methods section is pretty thorough, but some of the descriptions of the statistical/mathematical tools used to analyze the data are not really geared to a ‘virological’ audience.  If this work is to have more impact on this audience, it would be helpful to explain some of these tools in less jargony language.  For example, explaining what is a TG-ROC analysis and why do you use this could provide some context for people outside this area.

Response: References for TG-ROC analysis are provided, bur as suggested we briefly explained what TG-ROC analysis is about. “TG-ROC is a Microsoft-EXCEL spreadsheet template designed for selecting cut-off values in quantitative diagnostic tests at preselected accuracy level (e.g. 90 or 95% sensitivity and specificity.” Also please note that Fig. B should be Fig. A, Fig. A1 should be B1 – corrected accordingly in the revised manuscript.

3) p5, section 2.6: “laboratory” is misspelled

Response: Corrected.

4) p 11, section 3.5 (and/or in Discussion), it would be of interest to speculate on the relatively lower D-Se for cattle compared to goats, sheep.

Response: The following paragraph has been added to the text as per the Reviewer suggestion.

“The lower D-Se of rNP-based I-ELISA in cattle recorded in our study was also reported for a commercial competitive ELISA based on recombinant NP in Cameroonian cattle with D-Se ranging from 84.4% to 98.1% between different subpopulations tested [59]. The reason for lower D-Se remains unknown, but likely is due to higher cross-reactiveness of cattle sera”. 

5) Table 3 formatting is off and makes it very hard to read.

Response: Table 3 formatting improved as recommended.

6) Table 5 appears to lack any content

Response: We apologize for this, but it was uploaded on Viruses submission portal with original manuscript. Likely disappeared while our original submission was converted to Viruses’ manuscript format. Unfortunately the page on which the table should appear is not easy editable in the version of manuscript provided by Editorial Office of Viruses for revision. For this reason a pdf copy of the table is inserted to show its content.

7) Reference 34 in the bibliography should read “Wichgers Shreur, P.J….” instead of “Schreur, P.W.”

Response: Corrected to “Wichgers Schreur PJ” as per published list of authors.

Reviewer 3 Report

Summary:

The manuscript titled “Large-scale international validation of an indirect ELISA based on recombinant nucleocapsid protein of Rift Valley fever virus for the detection of IgG antibody in domestic ruminants” describes the validation of an indirect ELISA for detecting IgG antibodies against Rift Valley fever virus (RVFV) in livestock. The validation included a large sample set of sera from multiple countries including endemic and non-endemic areas. Additionally the authors compared the ELISA to virus neutralization tests (VNT) and provided optimized diagnostic accuracy for the test. The authors concluded that this ELISA is a valuable tool for RVFV surveillance globally. The assay is low cost, user-friendly and robust, which makes the assay quite valuable for low-income countries. This manuscript is generally well written and the evaluation of the assay is very thorough. I have the following comments:

The manuscript draft provided for review does not include line numbers. Therefore, comments will be directed to specific sections when possible. Additionally, throughout the manuscript there are extra spaces between the sentence period and the first word in the proceeding sentence or between two words in a sentence. Please carefully review the text for the extra spaces.  

ABSTRACT: Please elaborate on the statement “Practically simple incubation”. Simpler compared to what other method? How does one define practically simple?

INTRODUCTION

In the third paragraph, the sentence starting with “RVFV is a negative…” a period is missing after Phenuiviridae.

In the fourth paragraph, please elaborate what “using inactivated specimens before testing” means.

In the fifth paragraph, the indentation of the first word of the paragraph does not align with other paragraphs.

In the seventh paragraph, the discussion of recombinant antigen technology for RVFV detection is good but could be expanded. The authors discuss the NP and NSs antigens as DIVA compatible targets for diagnostics. This is a very important topic especially for RVFV where there are strict international trade regulations. Because of the restrictions there has been a push for developing multiplexing technologies for antibody detection. There is no discussion of these newer multiplexing technologies and their pros/cons compared to ELISAs. Please include a discussion on multiplexing serological assays for RVFV in relation to ELISA platforms as well as expand the importance of DIVA compatible assays.

RESULTS

Table 1: This table is nicely organized and formatted. Easy to read.

Figure 1: The manuscript appears to have several figure formatting issues. Graphs A and A1 are not in alignment with the rest of the graphs.

Section 3.2: In the sentence “ The lowest number of…” appears to have a random period in it after mean+3SD.

Figure 4: In the figure caption the distribution of IgG ELISA PP value is depicted in “blue”. Will this manuscript be printed in color? If not, consider changing this to gray like Figure 3.

Table 2: The rows are not in alignment.

Table 3: The columns are not in alignment. This table needs a lot of formatting work to improve clarity

Table 6: Minor formatting is needed for column titles. Some titles are bolded text but some are not.

Author Response

Reviewer 3 - We thank  for valuable comments.

Comments and Suggestions for Authors

The manuscript titled “Large-scale international validation of an indirect ELISA based on recombinant nucleocapsid protein of Rift Valley fever virus for the detection of IgG antibody in domestic ruminants” describes the validation of an indirect ELISA for detecting IgG antibodies against Rift Valley fever virus (RVFV) in livestock. The validation included a large sample set of sera from multiple countries including endemic and non-endemic areas. Additionally the authors compared the ELISA to virus neutralization tests (VNT) and provided optimized diagnostic accuracy for the test. The authors concluded that this ELISA is a valuable tool for RVFV surveillance globally. The assay is low cost, user-friendly and robust, which makes the assay quite valuable for low-income countries. This manuscript is generally well written and the evaluation of the assay is very thorough. I have the following comments:

The manuscript draft provided for review does not include line numbers. Therefore, comments will be directed to specific sections when possible. Additionally, throughout the manuscript there are extra spaces between the sentence period and the first word in the proceeding sentence or between two words in a sentence. Please carefully review the text for the extra spaces.  

Response: Line numbers included as requested. Text reviewed to remove extra spaces.

ABSTRACT: Please elaborate on the statement “Practically simple incubation”. Simpler compared to what other method? How does one define practically simple?

Response: “Practically simple incubation” changed to “Standard incubation”

INTRODUCTION

In the third paragraph, the sentence starting with “RVFV is a negative…” a period is missing after Phenuiviridae.

Response: Corrected.

In the fourth paragraph, please elaborate what “using inactivated specimens before testing” means.

Response: For more clarity it has been changed as follows: “A sandwich ELISA was developed for detection of nucleocapsid protein of RVFV in various clinical specimens.”

In the fifth paragraph, the indentation of the first word of the paragraph does not align with other paragraphs.

Response: Corrected.

In the seventh paragraph, the discussion of recombinant antigen technology for RVFV detection is good but could be expanded. The authors discuss the NP and NSs antigens as DIVA compatible targets for diagnostics. This is a very important topic especially for RVFV where there are strict international trade regulations. Because of the restrictions there has been a push for developing multiplexing technologies for antibody detection. There is no discussion of these newer multiplexing technologies and their pros/cons compared to ELISAs. Please include a discussion on multiplexing serological assays for RVFV in relation to ELISA platforms as well as expand the importance of DIVA compatible assays.

Response: Expended as recommended. And two relevant references added.

“A multiplex fluorescence microsphere immunoassay (FMIA) was developed to detect IgM and IgG antibodies in ruminant sera to RVFV structural and non-structural proteins. Preliminary results demonstrate the potential of FMIA diagnostic platform for development of diagnostic tests that can be used to differentiate vaccinated from infected animals and for simultaneous differential diagnosis of several abortive and zoonotic pathogens.”

1.Hossain, M.M.; Wilson, W.C.; Faburay, B.; Richt, J.; McVey, D.S.; Rowland, R.R. Multiplex detection of IgG and IgM to Rift Valley fever virus nucleoprotein, nonstructural proteins, and glycoprotein in ruminants. Vector Borne Zoonotic Dis. 2016, 16, 550-557.

2.Lindahl, J.F.; Ragan, I.K.; Rowland, R.R.; Wainaina, M.; Mbotha, D.; Wilson, W. A multiplex fluorescence microsphere immunoassay for increased understanding of Rift Valley fever immune responses in ruminants in Kenya. J. Virol. Methods 2019, 269, 70-76.

RESULTS

 Table 1: This table is nicely organized and formatted. Easy to read.

Response: Thank you for this comment.

Figure 1: The manuscript appears to have several figure formatting issues. Graphs A and A1 are not in alignment with the rest of the graphs.

Response: We apologize for this, but formatting likely changed after uploading on Viruses submission portal. Formatting figures improved, including alignment of graphs A and A1 in the revised manuscript. Also please note that Fig. B should be Fig. A, Fig. A1 should be B1 – corrected accordingly in the revised manuscript.

Section 3.2: In the sentence “ The lowest number of…” appears to have a random period in it after mean+3SD.

Response: Corrected throughout the text.

Figure 4: In the figure caption the distribution of IgG ELISA PP value is depicted in “blue”. Will this manuscript be printed in color? If not, consider changing this to gray like Figure 3.

Response: It was changed to gray like in Figure 3.

Table 2: The rows are not in alignment.

Response: We apologize for this, but formatting likely changed after uploading on Viruses submission portal. Corrected as recommended.

Table 3: The columns are not in alignment. This table needs a lot of formatting work to improve clarity

Response: We apologize for this, but formatting likely changed after uploading on Viruses submission portal. Corrected as recommended.

Table 6: Minor formatting is needed for column titles. Some titles are bolded text but some are not.

Response: Done as recommended.

Round 2

Reviewer 1 Report

The authors have responded to the reviewer suggestions and comments.